# Albumin Submicron Particles with Entrapped Riboflavin—Fabrication and Characterization

**DOI:** 10.3390/nano9030482

**Published:** 2019-03-25

**Authors:** Nittiya Suwannasom, Kathrin Smuda, Chiraphat Kloypan, Waraporn Kaewprayoon, Nuttakorn Baisaeng, Ausanai Prapan, Saranya Chaiwaree, Radostina Georgieva, Hans Bäumler

**Affiliations:** 1Institute of Transfusion Medicine, Charité-Universitätsmedizin Berlin, 10117 Berlin, Germany; Nittiya.Suwannasom@charite.de (N.S.); Kathrin.smuda@charite.de (K.S.); chiraphat.kloypan@charite.de (C.K.); Waraporn.Kaewprayoon@charite.de (W.K.); Ausanai.Prapan@charite.de (A.P.); Saranya.Chaiwaree@charite.de (S.C.); radostina.georgieva@charite.de (R.G.); 2School of Medical Sciences, University of Phayao, Phayao 56000, Thailand; 3School of Allied Health Sciences, University of Phayao, Phayao 56000, Thailand; 4Faculty of Pharmacy, Payap University, Chiang Mai 50000, Thailand; 5School of Pharmaceutical Sciences, University of Phayao, Phayao 56000, Thailand; patchateeya@yahoo.com; 6Faculty of Allied Health Sciences, Naresuan University, Phitsanulok 65000, Thailand; 7Department of Medical Physics, Biophysics and Radiology, Medical Faculty, Trakia University, 6000 Stara Zagora, Bulgaria

**Keywords:** riboflavin, immobilization, biopolymer, CCD-technique

## Abstract

Although riboflavin (RF) belongs to the water-soluble vitamins of group B, its solubility is low. Therefore, the application of micro-formulations may help to overcome this limiting factor for the delivery of RF. In this study we immobilized RF in newly developed albumin submicron particles prepared using the Co-precipitation Crosslinking Dissolution technique (CCD-technique) of manganese chloride and sodium carbonate in the presence of human serum albumin (HSA) and RF. The resulting RF containing HSA particles (RF-HSA-MPs) showed a narrow size distribution in the range of 0.9 to 1 μm, uniform peanut-like morphology, and a zeta-potential of −15 mV. In vitro release studies represented biphasic release profiles of RF in a phosphate buffered saline (PBS) pH 7.4 and a cell culture medium (RPMI) 1640 medium over a prolonged period. Hemolysis, platelet activation, and phagocytosis assays revealed a good hemocompatibility of RF-HSA-MPs.

## 1. Introduction

Riboflavin (RF), also known as vitamin B2, is a partially water-soluble vitamin that belongs to the group of flavoenzymes which catalyze oxidation-reduction reactions [1]. It is intrinsically fluorescent and has been used as modern drug [2]. It has been reported that RF has in vivo anti-metastatic properties in melanoma [3]. Several studies have shown that RF may also have antioxidant and anti-inflammatory effects [4,5]. Protective properties against cancer were shown in connection with co-enzyme Q10, RF, and niacin in tamoxifen-treated postmenopausal breast cancer patients [6]. RF has also been useful in photodynamic therapy (PDT). Because of its photosensitizing characteristics, it has a wide range of biological actions, such as inducing apoptosis in leukemia [7] and reducing the progression of prostate cancer cells [8], renal cancer cells [4], and melanoma [3]. Moreover, irradiated RF has been used to inactivate pathogens in blood transfusions [9] and it has the stabilized the corneal collagen crosslinking in keratoconus treatment [10].

RF is required in many oxidation-reduction reactions, and therefore RF deficiency may affect many systems [1]. RF is considered to be one of the most common vitamins with a deficiency in people of developing countries, particularly the countries where rice is their staple food. Consequently, a long-term use of RF supplement is required. Although it belongs to the water-soluble vitamins of group B, its solubility is about 2.65 × 10^−5^ mol/L^−1^ [11]. Therefore, micro-formulations based on hydrophobic interactions between RF and human serum albumin (HSA) may apply to overcome this limiting factor and to increase the therapeutic efficiency of the RF photosensitizer in cancer therapy [12,13].

The immobilization of compounds is a promising strategy for the improvement of stability, solubility, and biological activity through compound capture by carbonate microspheres in the process of their formation (Co-precipitation). The Co-precipitation Crosslinking Dissolution technique (CCD-technique) resulted in the fabrication of biopolymer particles using the precipitation of MnCl_2_ and Na_2_CO_3_ in the presence of a biopolymer solutions [14,15]. In the case of proteins, we used glutaraldehyde to crosslink the proteins in the MnCO_3_ template. The concentration was very low (<0.1%) and the final particles did not contain free aldehyde groups. Therefore, no toxic effects could be found [15]. The uniform peanut-like submicron particles were produced with a relatively high protein entrapment efficiency and a narrow distribution of around 700 nm. These carbonate particles could be easily loaded with bioactive compounds (e.g., enzyme) during their preparation [16,17,18,19]. The particle size and shape could be altered by adjusting the experimental conditions such as pH, choice of salt and/or salt concentration, temperature, and rate of mixing the solutions. This technique becomes increasingly interesting due to the high drug-loading capacity of the carbonate particles, the ease of preparation by simply mixing two starting solutions under mild conditions, and the complete dissolution of the carbonate template using EDTA at a neutral pH.

Micro- and nanoparticles made of human serum albumin (HSA) are an attractive alternative to synthetic polymers for use in the field of medicine and drug delivery due to their high binding capacity to both hydrophobic and hydrophilic drugs. Albumin nanoparticles showed the benefits of biocompatibility, biodegradability, non-toxic and non-immunogenic properties, thus avoiding inflammatory responses [20]. The HSA-based nanoparticles have been employed to deliver a variety of drugs such as brucine [21] and paclitaxel [22]. Various methods have been developed for the preparation of albumin particles such as desolvation/coacervation [23], emulsification [24], thermal gelation [25], nano spray drying [26], and self-assembly techniques [27] as well as co-precipitation which is used in our studies presented here.

For our investigations, RF served as model substance to demonstrate that more or less hydrophobic small molecules can be loaded into protein submicron particles using the CCD-technique. Additionally, the release of RF in vitro was studied in a phosphate buffered saline (PBS) and a cell culture medium (RPMI). Finally, we investigated the hemocompatibility of the RF containing HSA particles (RF-HSA-MPs), which is important for their application as intra-venously administered drug carriers.

## 2. Materials and Methods

### 2.1. Materials

Riboflavin (RF, >98% purity), glutaraldehyde (GA), fluorescein isothiocyanate (FITC), manganese chloride tetra-hydrate (MnCl_2_·4H_2_O), sodium carbonate (Na_2_CO_3_), phosphate buffered saline (PBS) pH 7.4, glycine, and sodium borohydride (NaBH_4_) were purchased from Sigma-Aldrich (Munich, Germany). Ethylene diamine tetra-acetic acid (EDTA) was purchased from Fluka (Buchs, Switzerland). Ampuwa^®^ (aqua ad injectable) and sterile 0.9% NaCl solution was purchased from Fresenius Kabi Deutschland GmbH (Bad Homburg, Germany). NaOH and DMSO were purchased from Carl Roth GmbH, Karlsruhe, Germany. Human serum albumin solution 20% was purchased from Grifols Deutschland GmbH (Frankfurt a.M., Germany). A Phagotest™ kit was purchased from Glycotope Biotechnology GmbH, Berlin, Germany.

### 2.2. Fabrication and Characterization of RF-HSA-MPs

#### 2.2.1. Fabrication of RF-HSA-MPs Particles

As RF is slightly soluble in water, a stock solution of 50 mM RF was prepared by dissolving it in 100% DMSO. The RF stock solution was protected from light by aluminum foil to prevent photo-degradation.

The RF-HSA-MPs were fabricated using a modified protocol based on the previously described CCD-technique [17,18]. Briefly, 20 mL of MnCl_2_ solution containing 10 mM RF and 10 mg/mL HSA were mixed in a 100 mL beaker for 1 min. Then 20 mL of Na_2_CO_3_ were added rapidly under vigorous stirring (Bibby Scientific CB161 Magnetic Stirrer, level 3) for 30 s at room temperature (final concentrations of RF and HSA were 5 mM (≈1.9 mg/mL) and 80 µM (5 mg/mL), respectively). The final concentration of MnCl_2_/Na_2_CO_3_ varied from 0.0625 to 0.25 M with a constant RF solution and HSA concentration. The hybrid particles obtained were separated by centrifugation at 3000× *g* for 3 min and washed twice with a 0.9% NaCl solution. The particles were suspended in a GA solution (final concentration 0.1%) and incubated at room temperature for 1 h, followed by centrifugation at 3000× *g* for 3 min. The remaining unbound aldehyde groups of GA in the particles were quenched using 0.08 M glycine and 0.625 mg/mL NaBH_4_, and the MnCO_3_ template was subsequently removed by treatment with EDTA solution (0.25 M, pH 7.4) at room temperature for 30 min. Finally, the resulting particles were centrifuged, washed until the washing solution became colorless, and resuspended in Ampuwa^®^ for further use. The fabrication scheme of the submicron particles is shown in Figure 1.

HSA particles with 4 mL DMSO without RF (HSA-MPs) were prepared following the same procedures and used as a control.

The amount of RF or HSA entrapped in the particles was determined as the difference between the total RF (RF_t_) or HSA (P_t_) amount added and the free non-entrapped RF (RF_f_) or HSA (P_f_) amount in the supernatants after co-precipitation and after each washing step. The RF concentration was determined spectroscopically measuring the absorbance of the supernatants at 445 nm with a microplate reader (PowerWave 340, BioTek Instruments GmbH, Bad Friedrichshall, Germany). The protein concentration was determined using a Coomassie Plus (Bradford) Assay Kit (Thermo Fisher Scientific, Waltham, IL, USA) with an absorbance measurement at 595 nm.

#### 2.2.2. Size, Zeta-Potential and Morphology of the HSA-MPs and RF-HSA-MPs

The size, polydispersity index, and zeta potential of the obtained particles were measured using a Zetasizer Nano ZS instrument (Malvern Instruments Ltd., Malvern, UK) at 25 °C. The particles were dispersed in PBS pH 7.4 and taken in a clear disposable zeta cell for zeta-potential measurement and in a plastic disposable cuvette for particle size measurement. Additionally, the particles were imaged using a confocal microscope (CLSM ZeissLSM 510 meta, Zeiss MicroImaging GmbH, Jena, Germany) and the size was assessed from the obtained images using the ImageJ-1 software (NIH, Bethesda, MD, USA).

The morphology of HSA-MPs and RF-HSA-MPs was investigated using an atomic force microscopy (AFM) in taping mode and a Nanoscope III Multimode AFM (Digital Instrument Inc., Santa Barbara, CA, USA). The samples were prepared on a freshly cleaved mica substrate pretreated with polyethylene imine (Mw 25 kDa, 1 mM for 20 min) by applying a drop of diluted particle suspension. The substrate was then rinsed with deionized water and dried under a gentle stream of nitrogen. The scans of the particles were first performed in the dry state, followed by the addition of a drop of deionized water and a scan in the wet state. For the scans in air (dry state) micro-lithographed tips on silicon nitride (Si_3_N_4_) cantilevers with a spring constant of 42 N/m and a resonance frequency of 300 kHz (Olympus Corporation, Tokyo, Japan) were used. Cantilevers with a spring constant of 3 N/m and a resonance frequency of 75 kHz (Budget Sensors, Innovative Solutions Bulgaria Ltd., Sofia, Bulgaria) were used for the measurements in the wet state. The Nanoscope software was used to record and analyze the obtained images.

#### 2.2.3. Intrinsic Fluorescence of the HSA-MPs and RF-HSA-MPs

The HSA-MPs and the RF-HSA-MPs were observed using a confocal laser scanning microscope (CLSM; ZeissLSM 510 meta, Zeiss MicroImaging GmbH, Jena, Germany) equipped with a 100× oil immersion objective (a numerical aperture of 1.3). Images of the samples were prepared in transmission and fluorescence mode with fluorescence excitation at 488 nm and a 505 nm long pass emission filter. The same settings were used for the imaging of the particles prepared with and without RF. Additionally, the particles were mounted on a glass slide using DakoCytomation fluorescent mounting medium and visualized using an Axio Observer (Zeiss, Göttingen, Germany). The fluorescence intensity was recorded at an excitation wavelength of 480 nm and an emission wavelength of 535 nm. A Zeiss filter cube no. 9 was used for fluorescence microscopy (EX 450–490, BS 510, EM LP 515).

The distribution of the fluorescence intensity inside the populations of the HSA-MPs and RF-HSA-MPs was analyzed using a flow cytometry (FACS-Canto II, Becton and Dickinson, Franklin Lakes City, NJ, USA) after diluting the samples with PBS at a ratio of 1:40 [28]. The performance of the flow cytometer was checked regularly using Cytometer Setup and Tracking Beads (BD Biosciences, Franklin Lakes, NJ, USA) to ensure the accuracy and precision of the measurements. A total of 10,000 events of particles were recorded from each sample. Subsequently, the fluorescence of the particles was determined in the PE-A channel as the relative median fluorescence intensity (RFI). The data were analyzed using the FlowJo v10 software (Tree Star, Ashland, OR, USA).

### 2.3. In Vitro Release of RF from the RF-HSA-MPs

For the release studies, 2.5 mL of 16% (*v*/*v*) RF-HSA-MPs suspension were transferred into a dialysis membrane sleeve (Cellu Sep T3, MWCO 12,000–14,000, Creative BioMart, Shirly, NY, USA) and sealed at both ends after adding 1 mL release media (0.1 M PBS pH 7.4 or RPMI 1640 medium supplemented with 10% fetal bovine serum (FBS) and 1% PenStrep to mimic the biological environment). The dialyzer was then introduced into a 25 mL glass cylinder containing 9 mL of release media (0.1 M PBS pH 7.4 or RPMI 1640 medium), which was stirred continuously at 100 rpm using a magnetic stir bar at room temperature. The samples were removed from any light because of the light sensitivity of RF. The RF-release was assessed intermittently by sampling (400 µL) the contents of the outer media and replacing this with an equal volume of fresh PBS pH 7.4 or RPMI 1640 medium immediately after sampling, correspondingly. The amount of released RF was measured at a wavelength of 445 nm using an UV-vis spectrophotometer (Hitachi U2800, Hitachi High-Technologies Corporation, Kreefeld, Germany).

The release profiles of RF in PBS pH 7.4 and in RPMI 1640 medium were displayed as time dependency for the remaining RF concentration in the RF-HSA-MPs and fitted with the release model of Pappas [29], Equation (1) was used:m (t)/m (∞) = k_1_t^n^ + k_2_t^2n^(1)where m (t)/m (∞) is the cumulative drug release, t is the release time (in hours), the first term of the right side is the Fickian contribution (F), the second term is the Case-II relaxational contribution (R) that reflects the structural and geometric characteristics of the MPs, and n is a diffusional exponent that is characteristic for the controlled release of the loaded drug [30,31].

### 2.4. Hemocompatibility of HSA-MPs and RF-HSA MPs

Freshly withdrawn venous blood was collected from healthy volunteers and anticoagulated using lithium heparin (368494, BD Vacutainers) or into sodium citrate (366575, BD Vacutainer). The blood samples were collected at the Charité—Universitätsmedizin Berlin (# EA1/137/14) and all donors provided written informed consent. The blood samples were mixed gently (but thoroughly) to ensure adequate mixing with the anticoagulant immediately after blood collection. All samples were processed within 2 h of blood collection.

#### 2.4.1. Hemolysis Test

The hemolytic activity was determined on the release of hemoglobin from damaged erythrocytes in vitro. Human heparinized blood was washed with PBS by centrifugation at 3000× *g* for 5 min to isolate the red blood cells (RBCs). The RBCs were further washed until a colorless pellet was obtained and then diluted to achieve a cell suspension with a volume concentration of 2% in PBS. Then, 0.5 mL of the 0.5%, 1%, and 2% diluted RBCs suspension was mixed with 0.5 mL of 2% HSA-MPs, RF-HSA-MPs, double distilled water as the positive (PC) or PBS as a negative (NC) control. After incubation at 37 °C for 3 h and centrifugation at 3000× *g* for 5 min, the supernatants were transferred carefully to a 96-well plate and the absorbance was measured using a microplate reader at 545 nm. The degree of hemolysis was calculated as the hemolytic ratio (HR) using Equation (2):HR% = (*A_test_* − *A_NC_*)/(*A_PC_* − *A_NC_*) × 100%(2)
where $A_{test}$ is the absorbance of the tested sample, $A_{NC}$ is the absorbance of the negative control in PBS and $A_{PC}$ is the absorbance of the positive control in distilled water.

#### 2.4.2. Phagocytosis Test

The interaction of the HSA-MPs and RF-HSA-MPs with the blood leukocytes was evaluated in vitro in human whole blood using a commercial Phagotest kit (Glycotope-Biotechnology GmbH, Heidelberg, Germany). Manufacturer’s instructions were partially modified: all reactions were performed with half of the volume (50 µL instead of 100 µL), lysing solution was changed to ammonium chloride lysing solution (155 mM NH_4_Cl, 12 mM NaHCO_3_, 0.1 mM EDTA), and DNA was not stained. To put it briefly, 10 µL of 2 × 10^11^ per mL, RF-HSA-MPs, HSA-MPs, were added into 50 µL heparinized whole blood and carefully mixed. For the negative control 10 µL of PBS were added to 50 µL blood and for the positive control (functional test of the granulocytes and monocytes in the blood) 10 µL of 2 × 10^11^ FITC-labeled opsonized *E. coli* (positive control) were applied. The samples were incubated at 37 °C for 30 min (PBS and FITC-labeled opsonized *E. coli* were incubated for 10 min). The control samples remained on ice. At the end of the incubation period, all samples were placed in the ice-bath. A quenching solution was added and washed with ice-cold PBS. The erythrocytes were lysed with ammonium chloride solution for 15 min. The cells were washed twice and re-suspended in ice-cold PBS. The percentage of granulocytes and monocytes exhibiting phagocytosis was determined using a flow cytometer (BD FACS Canto II, Franklin Lakes, NJ, USA).

#### 2.4.3. Platelet Activation Test

The effect of the HSA-MPs and RF-HSA-MPs on the function of the blood platelets was tested in a platelet-rich plasma (PRP). The PRP was isolated from human whole blood anticoagulated with sodium citrate by centrifugation at 150× *g* for 15 min and used immediately. Then the HSA-MPs and RF-HSA-MPs were added to the PRP at a final ratio of 5 particles per 1 platelet, carefully mixed and incubated in a water bath at 37 °C for 30 min. A negative control was prepared adding the same volume of PBS instead of particle suspension. To induce activation and aggregation of the platelets, the pre-incubated PRP samples were treated with 0.5 mg/mL of arachidonic acid or 0.018 mg/mL of epinephrine (Mölab, Langenfeld, Germany) at 37 °C for 30 min. An ABX Micros 60 hematology analyzer (Horiba Europe GmbH) was used to detect the platelet number in the samples before and after incubation. Finally, the platelets were stained with APC-mouse anti-human CD41a and Alexa Fluor^®^ 488-mouse anti-human CD62p (p-selectin), kept in the dark for 20 min, and fixed with 500 µL of a fixative solution (0.5% paraformaldehyde in PBS) to each test tube to stop the reactions. The expression of the platelet activation marker CD62P and the constitutively present platelet marker CD42b were analyzed using a flow cytometry (BD FACS Canto II).

## 3. Results and Discussion

### 3.1. Fabrication and Characterization of RF-HSA-MPs

#### 3.1.1. HSA and RF Content, Size, Zeta-Potential and Morphology

In previous studies by our group, it has been shown that the co-precipitation technique is much more effective for the protein entrapment than the absorption onto the carbonate particles [32,33]. Moreover, it was found that the entrapment of proteins using the MnCO_3_ template was higher than that of the CaCO_3_ template [14,15]. The encapsulation efficiency was attributed to the electrostatic attraction between negatively charged proteins and more positively charged MnCO_3_ as well as to the stronger affinity of Mn^2+^ to proteins and in particular to HSA [34].

However, the addition of low molecular weight compounds into polymeric particles and capsules still remains a challenge. In our study we used RF as a model to investigate the potential of the CCD-technique to deliver carrier systems for low molecular weight drugs with poor water solubility. The weak water-soluble RF was added together with HSA via the CCD-technique as shown in Figure 1. To achieve this RF was already added during the first step of the particle preparation, the co-precipitation, together with HSA. It had been previously shown that RF interacts with albumin through adsorption on the tryptophan residues via hydrophobic interactions [12,13], which was expected to support the RF entrapment into the HSA-MPs.

The co-precipitation was performed at the previously optimized concentration of MnCl_2_ and Na_2_CO_3_ for the entrapment of HSA (0.125 M). The average amounts of entrapped HSA and RF particles under these preparation conditions were 2.9 ± 0.8 mg and 2.5 ± 0.5 mg per mL, respectively. This means that in a particle suspension with a volume concentration of 10%, the RF concentration will be roughly 290 µg/mL which is over four times higher than the solubility of RF in water at 20 °C (70 µg/mL, GESTIS—materials database).

On CLSM images the HSA-MPs and RF-HSA-MPs exhibited a size between 0.9 and 1.1 µm with an average long diameter of 1 μm. These values correlated well with the measurements using the dynamic light scattering (Zetasizer Nano ZS, Malvern Instruments Ltd., Malvern, UK) which delivered values of 1.04 ± 0.15 µm. There were no significant differences found between HSA-MPs and RF-HSA-MPs. Under the conditions chosen for this study, which were a rapid mixing of all compound at room temperature, the size of the particles was highly reproducible.

The main factors that determine the size of the MnCO_3_ particles were the concentrations of manganese and carbonate ions, the flow rate of the solutions during mixing, and the temperature. Particular variations of these parameters are needed for controlling size and shape of the particles in co-precipitation reactions [35]. The mixing of MnSO_4_ and NH_4_HCO_3_ has been widely employed to prepare manganese carbonate particles and used as scarified templates for the assembly of polyelectrolyte multilayers via layer-by-layer (LBL) self-assembly technique. Micron MnCO_3_ crystals with different size distributions varying from 1 to 10 µm have been synthesized at low concentration ratios of MnSO_4_/NH_4_HCO_3_ with long precipitation times and additional solvents at high temperature [36,37,38,39,40]. Subsequently, the manganese carbonate core was dissolved in HCl at low pH. In this study, MnCO_3_ was synthesized by a co-precipitation method using MnCl_2_ and Na_2_CO_3_ as the manganese and carbonate source, respectively. The precipitation was completed very fast, at room temperature with high salt concentration, and the dissolution was completed with EDTA at neutral pH. These conditions are suitable for the preparation of protein particles avoiding denaturation and preserving the function of the proteins.

The morphology of HSA-MPs and RF-HSA-MPs was analyzed using an AFM as shown in Figure 2. The shape of both kinds of particles was peanut-like. The long diameter measured for both kinds of particles varied between 780 and 890 nm without significant differences between them. The thickness of the particles was determined from the height profiles were 400 ± 45 nm, which corresponds to half of the long diameter. The addition of RF did not seem to interfere with the geometry of the particles.

The RF-HSA-MPs und HSA-MPs were further investigated with respect to their electrokinetic potential (zeta-potential). This parameter is important for the stability of a particle suspension, in particular for the behavior of the particles in biological fluids. Therefore, three measurements of the zeta-potential were conducted in PBS pH 7.4 (conductivity 17 mS/cm). Both HSA-MPs and RF-HSA-MPs exhibited zeta-potential of approximately −15 mV, which is a relatively high value at the high ionic strength of PBS. In water (conductivity 14 µS/cm) the zeta-potential was approximately −39 mV, which contributed to the high colloidal stability and absence of aggregation of the particles in a biologically relevant media.

#### 3.1.2. Intrinsic Fluorescence of the HSA-MPs and RF-HSA-MPs

Both HSA-MPs and RF-HSA-MPs could be detected in the fluorescence channels of the confocal microscope. A weak autofluorescence due to the GA crosslinking was observed in the HSA-MPs as seen in Figure 3(A1,A2), whereas the RF-HSA-MPs showed significantly stronger fluorescence due to the entrapped RF as seen in Figure 3(B1,B2).

More clearly the difference of the fluorescent emission is demonstrated in the 3D color surface map representing a single HSA-MP and RF-HSA-MP in Figure 3(A3,B3), respectively.

The higher value of the intrinsic fluorescence of the RF-HSA-MP confirms the successful entrapment of the drug into the particles. Additionally, the intrinsic fluorescence is very useful for tracking these particles when they interact with cells without the need for additional labeling.

### 3.2. In Vitro Release of RF from the RF-HSA-MPs

The investigation of the drug release was performed using a dialysis-bag diffusion method against PBS pH 7.4 as well as RPMI 1640 medium. The cut-off of the dialysis bag allowed the free diffusion of released RF through the semi-permeable membrane from the solution inside the dialysis bag to the outside following the concentration gradient. The results of the in vitro release of RF from RF-HSA-MPs are shown in Figure 4a. The decrease of the RF concentration remaining in the particle suspensions was followed for 80 h. It can be seen that the drug release profiles, in both investigated media, are bi-phasic with an initial burst release of approximately 7% in PBS and 12% in RPMI from the initial RF concentration during the first 2 to 3 h. Thereafter, the release rate decreased and a sustained release was observed until the end of the experiments. After 80 h the drug release remained 30% and 45% of the initial loading in PBS and in RPMI, respectively.

The release in the RPMI 1640 medium which contained varying amino acids and 10% calf serum albumin was significantly faster, probably due to the adsorption of the released RF by these compounds, which resulted in a clearance of the free RF from the solution. Consequently, the concentration of the freely dissolved RF remained lower in the RPMI as compared with the concentration of the freely dissolved RF in the PBS, which led to a faster release. A similar release behavior was shown for a hydrophobic anti-cancer drug from a micelle system. The release was accelerated in buffers containing albumin [41] due to the binding of the drug to the hydrophobic regions of albumin.

The release profiles were fitted using the model of Pappas Equation (2), which is suitable to describe bi-phasic controlled release of entrapped drugs from particles. The values calculated for K_1_ are larger than those calculated for K_2_ by more than one magnitude for both PBS and RPMI. This indicates that the release is dominated by the diffusion mechanism. In RMPI the domination by the Fickian diffusion is much stronger due to the greater RF concentration gradient between RF in the RF-HSA-MPs and the bulk RPMI-phase.

### 3.3. Hemocompatibility of RF-HSA MPs

#### 3.3.1. Hemolysis Test

Hemolysis tests were performed to assess the impact of HSA-MPs and RF-HSA-MPs on the membrane stability of human erythrocytes. The HSA-MPs and RF-HSA-MPs showed low hemolytic activity with the percentage of hemolysis in the range of 4–7% and in a dose-dependent manner as shown in Figure 5. Therefore, the HSA-MPs and RF-HSA-MPs did not cause strong hemolytic effects. However, according to criterion listed in the ASTM E2524-08 standard, more than 5% hemolysis indicates damage to RBCs caused by the test materials. This critical value was reached at particle concentration of 1% for both HSA-MPs and RF-HSA-MPs.

In general, size, surface charge, and surface area are key parameters that affect the hemolytic potential of particles. Negatively charged particles interact less with the negative charged cell surface than positively charged particles. Micron-sized particles are more likely to produce a lower level of hemolysis than smaller particles [42,43,44]. The increase in surface-to-volume ratio with the decrease in size of particles results in enlarged surface contact area and provides the chance for damage to take place to a cell membrane. This could explain the dose-dependent increase of hemolysis observed with the HSA-MPs and RF-HSA-MPs.

#### 3.3.2. Phagocytosis Test

The ability of the HSA-MPs and RF-HSA-MPs to induce phagocytic activity of granulocytes and monocytes in whole blood was analyzed using a standard phagocytosis kit. Representative results of these tests are shown in Figure 6. The fluorescence signal from HSA-MPs and RF-HSA-MPs was detected in the PE-A channel of the flow cytometer, and the FITC-labelled *E. coli* (used as a standard positive control for phagocytosis) was detected in the FITC-A channel. The three main populations of white blood cells were identified based on their forward scatter (FSC) and side scatter (SSC): granulocytes, monocytes, and lymphocytes (dot-plot Figure 6A). The histograms in Figure 6B,C represent the distribution of the fluorescence intensity within the population of HSA-MPs and RF-HSA-MPs in the PE-A-channel. The higher fluorescence emission of the RF-HSA-MPs is clearly visible in the shift of their histogram by one order of magnitude to higher fluorescence intensities. The analysis of the fluorescence distribution in the granulocyte and monocyte populations of the samples incubated at 37 °C with FITC-*E. coli* (Figure 6D,G) shows a strong right shift in the FITC channel due to the engulfment of the fluorescent bacteria. This was not the case in the samples incubated with HSA-MPs and RF-HSA-MPs (Figure 6E,I). The particles did not induce phagocytosis, and therefore their immunogenicity is low. Avoiding clearance by phagocytosis is very important in drug delivery systems using micro-particles and in many cases requires complicated and expensive surface modification of the drug carriers [45]. Therefore, our HSA-MPs and RF-HSA-MPs are very promising for use in applications for drug delivery systems.

#### 3.3.3. Platelet Activation Test

Further, platelet activation was determined by evaluating expression of CD62p (P-selectin) and CD42b platelet surface markers. Non-treated PRP (negative control) showed nearly 10% platelet activation (expression of CD62p) caused by sample handling and preparation. Incubation with agonists (arachidonic acid, epinephrine, and collagen) caused an increased expression of CD62p in the platelets confirming their functionality. The measurement of the CD42b/CD62p co-expression in platelet samples treated with the HSA-MPs or RF-HSA-MPs revealed that there was no effect on the CD62p expression in CD42b positive cells. This is comparable to the control sample. Together with agonist, HSA-MPs or RF-HSA-MPs did not induce a different behavior in the activation of the platelets in comparison with the samples treated with agonists only. Therefore, both HSA-MPs and RF-HSA-MPs did not activate the platelets and did not augment the platelet response to antagonists. Representative dot plots are shown in Figure 7B and summarized results of the platelet activation test are shown in Figure 7B.

## 4. Conclusions

In conclusion, we demonstrated that the encapsulation of a drug with a low molecular weight and low water-soluble macromolecule, RF, can be performed by capturing the growing MnCO_3_ particles together with HSA. The negatively charged particles can be produced with a narrow size distribution and diameters less than 1 μm. The release of RF from the particles exhibits bi-phasic profile with a dominating Fickian diffusion mechanism. These findings suggest that RF-HSA-MPs represent a compelling strategy for a long-term drug delivery system, and that the CCD-technique of incorporation is applicable to various biomolecules with different molecular weights. Taken together, with the investigation of the release of RF and the hemocompatibility, this work provides basic information for the production and application of HSA-based micro-particles as a drug carrier system.

## Figures and Tables

**Figure 1 nanomaterials-09-00482-f001:**
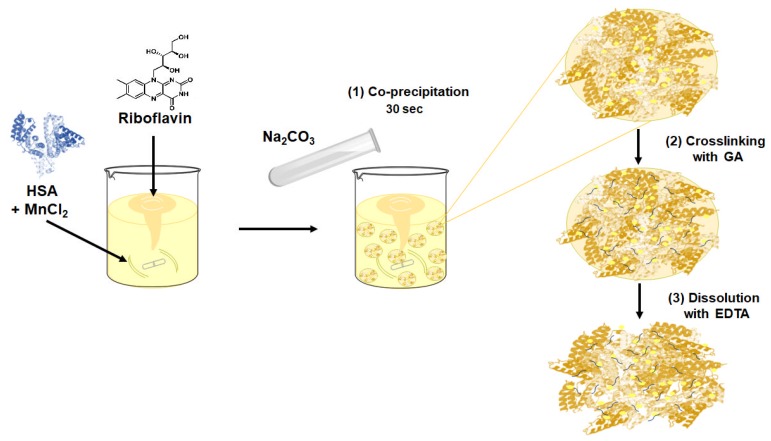
Scheme of the fabrication procedure for the submicron human serum albumin (HSA) particles containing riboflavin (RF), i.e., modified Co-precipitation Crosslinking Dissolution technique (CCD-technique).

**Figure 2 nanomaterials-09-00482-f002:**
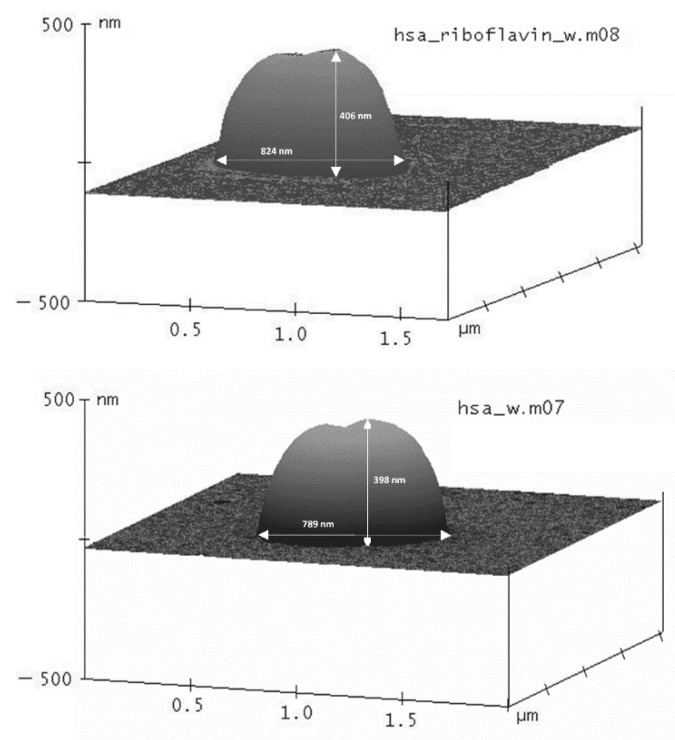
AFM images of RF containing HSA particles (RF-HSA-MPs) (**top**) and HSA particles with 4 mL DMSO without RF (HSA-MPs) (**bottom**) in three dimensional (3D)-mode. The size of the particles was determined from the height profiles in horizontal and vertical directions. The values included in the images are representative examples.

**Figure 3 nanomaterials-09-00482-f003:**
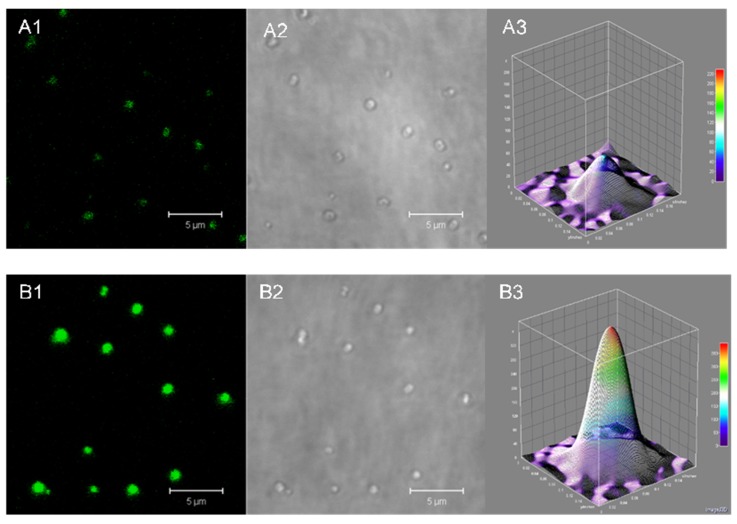
Analysis of intrinsic fluorescence in HSA-MPs and RF-HSA-MPs. Confocal micrograph of HSA-MPs in (**A1**) fluorescence mode and (**A2**) transmission mode, respectively and confocal micrograph of RF-HSA-MPs in fluorescence mode (**B1**) and transmission mode (**B2**), respectively. Fluorescence emission intensity in 3D color map surface images of (**A3**) HSA-MPs and (**B3**) RF-HSA-MPs at an excitation wavelength of 480 nm and an emission wavelength of 535 nm.

**Figure 4 nanomaterials-09-00482-f004:**
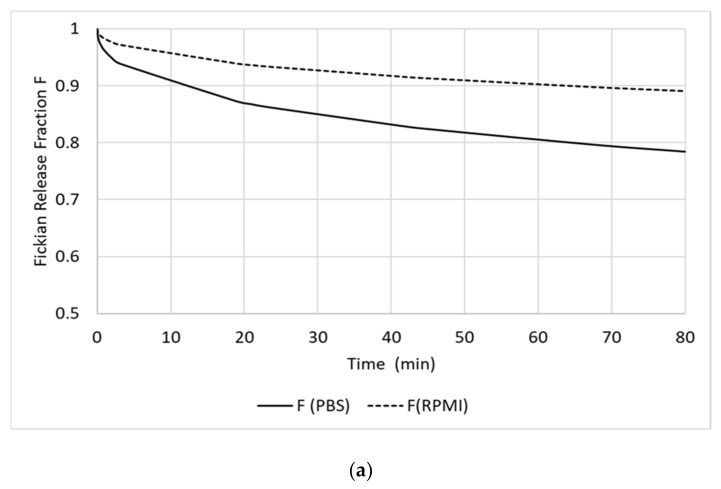
(**a**) Release profiles of RF in the phosphate buffered saline (PBS) pH 7.4 (●) and in the cell culture medium (RPMI) 1640 medium (♦) at room temperature calculated for the remaining RF concentration entrapped in the RF-HSA-MPs. The RF concentrations were fitted using the Pappas equation m (t)/m (∞) = k_1_t^n^ + k_2_t^2n^. Values are expressed as mean ± SD (n = 3). (**b**) Fickian release fraction (F) of RF-HSA-MPs in PBS and in RPMI. The release of RF in RPMI is much stronger in the domination by the Fickian diffusion, due to the greater RF concentration gradient between RF in the RF-HSA-MPs and the bulk RPMI-phase.

**Figure 5 nanomaterials-09-00482-f005:**
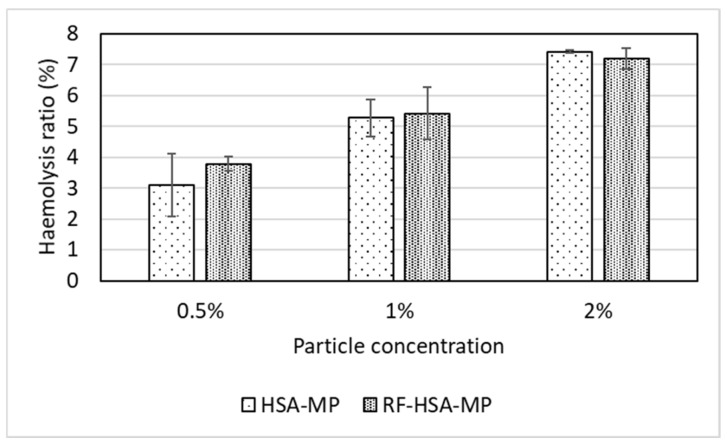
Hemolytic activity induced by HSA-MPs and RF-HSA-MPs for 3 h at 37 °C, concentration ranging from 0.5%, 1%, and 2%. Water and PBS served as positive (100%) and negative (0%) control, respectively. Data are presented as the mean percentage ± SD (n = 3).

**Figure 6 nanomaterials-09-00482-f006:**
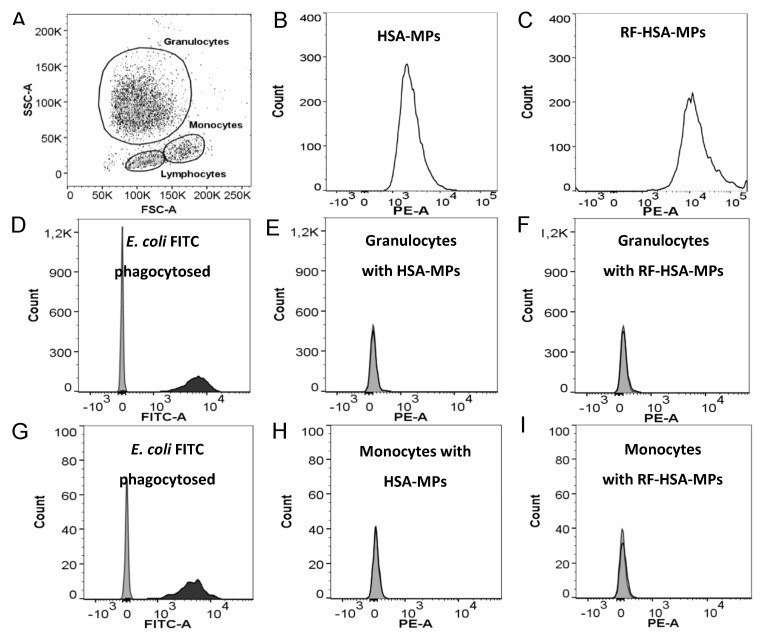
Phagocytosis assay showing (**A**) three groups of white blood cells identified by their forward scatter (FSC) and side scatter (SSC): granulocytes, monocytes, and lymphocytes. Histograms representing fluorescence intensity of (**B**) HSA-MPs and (**C**) RF-HSA-MPs in the PE-A channel. Phagocytosis of FITC-labeled opsonized *E. coli* in granulocytes (**D**) and monocytes (**G**) (grey area: negative controls incubated at 0 °C; black area: granulocytes and monocytes with phagocytosed FITC *E. coli*). Representative histograms of fluorescence intensity of monocytes and granulocytes after interacting with (**E**,**H**) HSA-MPs and (**F**,**I**) RF-HSA-MPs (grey area: negative controls incubated at 0 °C; black line: granulocytes and monocytes incubated at 37 °C MPs; k stands for kilo). Data are representative of n = 3 independent phagocytosis assays showing the same trends.

**Figure 7 nanomaterials-09-00482-f007:**
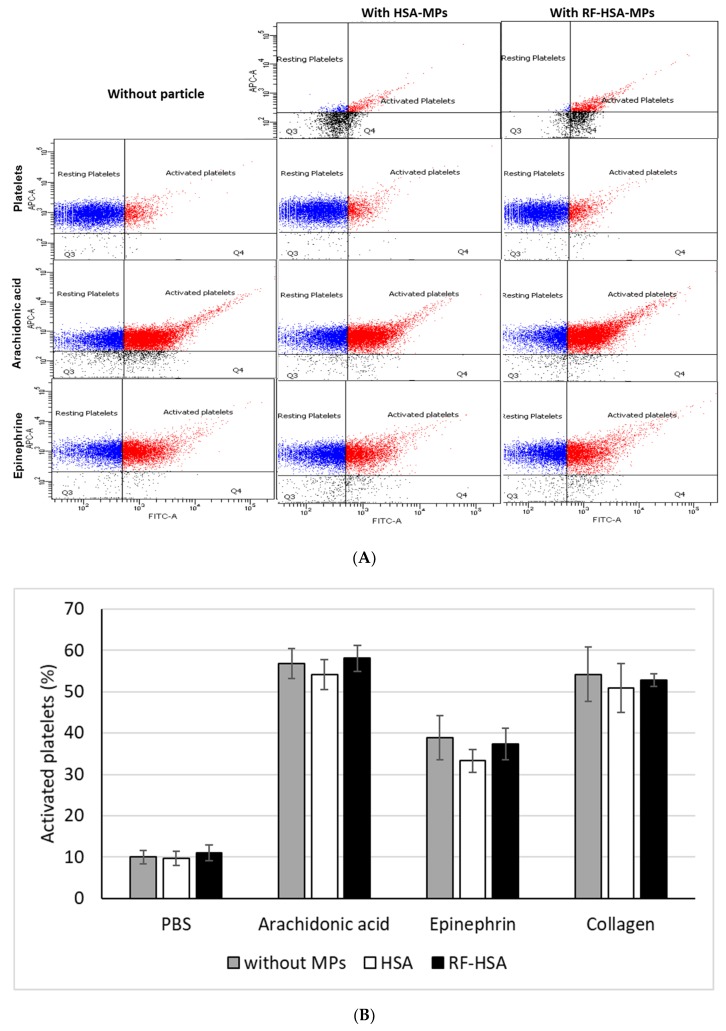
Platelet activation assay showing (**A**) representative dot plots in APC-A/FITC-A channel for HSA-MPs and RF-HSA-MPs (upper row); CD42b (APC-A)/CD62P (FITC-A) dot plots gated for platelets at rest and after stimulation with HSA-MPs and RF-HSA-MPs (second row); CD42b (APC-A)/CD62P (FITC-A) dot plots gated for platelets after stimulation with arachidonic acid, arachidonic acid and HSA-MPs, arachidonic acid and RF-HSA-MPs (third row); CD42b (APC-A)/CD62P (FITC-A) dot plots gated for platelets after stimulation with epinephrine, epinephrine and HSA-MPs, epinephrine and RF-HSA-MPs (lower row); (**B**) FACS analysis of MPs platelet activation measured by the determination of CD62p/CD42 co-expression. The presence of particles did not have an influence on platelet activation. The agonists (arachidonic acid, epinephrine, and collagen) induced platelet activation independent from the presence of particles. Data are representative of n = 4 independent platelet activations showing the same trends and are presented as mean ± SD (n = 4).

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
