# Peer review of "Albumin Submicron Particles with Entrapped Riboflavin—Fabrication and Characterization"

_nanomaterials, 2019, doi:10.3390/nano9030482_

Reviewer 1 Report

The manuscript submitted for evaluation deals with some encapsulation of riboflavin in micron-size materials of albumin type. The work is well conducted and the conclusions are adequate. The presentation of the manuscript is also good, although there is a mistake (lines 449-459 should be removed). The science behind is well known, and therefore there is a feeling of not very exciting results, but this can be considered overcome by the amount of data presented and the clarity of interpretation. The paper can be published as it is, with some minor corrections.

Author Response

Thank you for the comments. The text after References was removed.

Reviewer 2 Report

The authors describe preparations of particles for transport of riboflavin. The particles were prepared by coprecipitation crosslinking dissolution technique and characterized by DLS and AFM. Hemolysis and drug release are described.     

The preparation is not well described. No one could prepare particles according to the procedure. At least one complete protocol should be given. The text should be better controlled and mistakes corrected. Question is if all the figures are necessary. Some text in references should be removed.

Author Response

Thank you for the comments.

Since your comments arrived after the revision of the manuscript I ask you to read the revised manuscript.

The fabrication of particles was described in detail in the references 14-16. Therefore only a brief description was given. We hope you can accept it.

The mentioned text in the references was removed.

Reviewer 3 Report

The research paper reports submicron particles based on albumin for the encapsulation of riboflavin. Although the approach is interesting, this reviewer considers that the authors didn’t justify the interest of the formulation. The publication needs major revision and improvements at the level of rational and relevance of the work.

What is the route of delivery that is being intended? That must be clearly indicated.

The assays include testing hemocompatibility, which makes sense when proposing a system for intravenous delivery. However, that is not certainly the case, because of the size of particles of around 1 μm. Please address this aspect in the discussion and justify the rationale of the work on the introduction.

The presence of glutaraldehyde in the systems raises concern. Depending on the route of delivery that is selected, adequate cells lines should be used to test the cytotoxicity of the system.

In line 94, please indicate the used volume of HSA solution at concentration 10 mg/mL.

The zeta potential was determined in PBS, because the authors wanted to simulate the behaviour in contact with biological fluids. However, as the particles are produced in aqueous medium, characterising size and zeta potential in water is relevant for stability issues. Please include these data.

Figure 3 shows fluorescence in HAS microparticles, which the authors attribute to glutaraldehyde crosslinking. What about the fluorescence of tryptophan? Please comment on this.

The sentence “which is beneficial for the priming response of an effective therapeutic” in line 323 is misleading, please rewrite.

Sentence in lines 325-326 (“probably due to ….”) makes no sense, please revise this.

In line 334 it is said “In RPMI the predomination by the Fickian diffusion is much stronger due to the greater RF concentration gradient…”. Please discuss on the consequences of that considering the delivery on an in vivo organism and contextualising with the delivery route of selection.

Line 343, from legend of figure 4: “is much stronger predominated by the…” is not understandable. Please revise the sentence.

The authors must perform a very careful revision of the text in order to improve the English, to eliminate grammatical errors, typos and rewrite some misleading sentences. Some examples are given below, but all the text needs careful revision.

-          Lines 35-37: “…RF has … cancer-protective properties in connection with co-enzyme Q10, RF and niacin in… “ – something is repeated, sentence makes no sense.

-          Line 47: “…ranging from 2.65x10-5 mol.L-1…” – revise, ranging indicates that there is an interval and no interval is mentioned

-          Line 60: “This technique is increasingly gained interest…” – needs revision

-          Line 73: “model substance to demonstrate that more or less hydrophobic small molecules…” – needs revision

-          Line 236: “…direct adsorption of from…” – needs revision

-          Line 243: “… could to proteins and in particularly…” – needs revision

-          Line 265: “The main factors determining the size and of the…”

After the references, there is a portion of text that comes from the instructions for authors that needs to be eliminated.

Author Response

Thank you for your comments. We appreciate your hints and suggestions and included them into the revised manuscript. We used the tracking mode in Word and all changes are visible.

Additionally, we would like to answer your questions:

The assays include testing hemocompatibility, which makes sense when proposing a system for intravenous delivery. However, that is not certainly the case, because of the size of particles of around 1 μm. Please address this aspect in the discussion and justify the rationale of the work on the introduction.

Classical experiments of microcirculation demonstrate that particles in this size range can pass all capillaries. Our paper reference 15 (ACS Nano) clearly shows that these particles pass arterioles and the glomerulus of kidney.

The presence of glutaraldehyde in the systems raises concern. Depending on the route of delivery that is selected, adequate cells lines should be used to test the cytotoxicity of the system.

All previous experiments and analyses, which were performed for protein particles of different composition show that there are no free aldehyde groups. Cytotoxicity was not found (Ref 14-16, 18, 19 and  Ijad Kao, Yu Xiong, Axel Steffen, Kathrin Smuda, Lian Zhao, Radostina Georgieva, Hans Bäumler. Preclinical SAFETY investigations of submicron sized hemoglobin based oxygen carrier HBMP-700. Artificial Organs (2017) DOI:10.1111/aor.13071 (2018),42(5):549-559

Figure 3 shows fluorescence in HAS microparticles, which the authors attribute to glutaraldehyde crosslinking. What about the fluorescence of tryptophan? Please comment on this.

The tryptophan fluorescence is not visible in the image, because the excitation wavelength is 290 nm for tryptophan and the emission is in the range from 330 to 390 nm.

In line 334 it is said “In RPMI the predomination by the Fickian diffusion is much stronger due to the greater RF concentration gradient…”. Please discuss on the consequences of that considering the delivery on an in vivo organism and contextualising with the delivery route of selection.

The model is only applicable for diffusion experiments. That means, if the particles are endocytosed the entrapped drug diffuses due to the concentration gradient. In case of intravenously administration the particles will circulate and convection has to be considered. This is a completely different situation.

We hope that you can agree with our explanations.

Thank you.

Best regards

Hans

Round  2

Reviewer 2 Report

The fabrication of particles was described in detail in the references 14-16. Therefore only a brief description was given. We hope you can accept it.

- Best protocol should be given.

The mentioned text in the references was removed.

- No, it was not.

- English was not checked perfectly. For example, p.7, l.272 is the word dissoluted correct?

l. 278 ws?, l.287 thee? l.323 will remains? ...

Author Response

Sorry, something went wrong with the reference paragraph. Now it was deleted.

dissoluted as well as dissolved are possible. Up to now, we used "dissoluted" in all of our publications.

Thank you for your hint: l. 278 ws?, l.287 thee? l.323 will remains?

We corrected it now.

Reviewer 3 Report

The authors addressed almost all the concerns. It is the opinion of this reviewer that the question of glutaraldehyde toxicity should at least be addressed in the introduction, and the comments will be very helpful for the readership. 

The manuscript still needs revision because of several typos.

Author Response

We hope that we found all mistypings etc. now.

Concerning glutaraldehyd. We agree with you and included a sentence in the introduction. We cannot expect that all readers are familiar with effects of very low concentrations of GA.

Thank you.